# Evaluation of Six Years of Appropriateness Level of Blood Transfusion in a Pediatric Ward

**DOI:** 10.3390/ijerph20031700

**Published:** 2023-01-17

**Authors:** Pier Mario Perrone, Gregorio Paolo Milani, Rosa Maria Dellepiane, Antonella Petaccia, Daniele Prati, Carlo Agostoni, Paola Giovanna Marchisio, Silvana Castaldi

**Affiliations:** 1Department Biomedical Sciences for Health, University of Milan, 20122 Milan, Italy; 2Fondazione IRCCS Ca’ Granda Ospedale Maggiore Policlinico, 20122 Milan, Italy; 3Department of Clinical Sciences and Community Health, University of Milan, 20122 Milan, Italy

**Keywords:** blood transfusions, pharmacological appropriateness, pharmacoepidemiology, appropriate prescribing, public health

## Abstract

Background: Blood transfusion can be considered as a life-saving treatment and is a primary health management topic. This study aims to assess the appropriateness of blood transfusion performed in a large tertiary hospital in Italy. Methods: a multispecialist team composed oof hematologists, public health experts and pediatricians analyzed blood transfusions performed between 2018 and 2022 in the pediatric wards comparing the appropriateness with the available NHS guidelines available. Patients’ characteristics, clinical features and blood component’s data were collected and analyzed. Results: considering 147 blood transfusions performed in 2018–2022, only eight (5.4%) were performed according to guidelines, while 98 (66.7%) were driven by clinicians’ expertise, especially for anemia in genetic syndromes (30) (20.5%) and autoimmune diseases (20) (13.6%). Thirty-nine (26.5%) transfusions could be considered as inappropriate, while two (1.4%) blood packs were never been transfused after being requested. Conclusions: This analysis is one of the first performed to assess the appropriateness of blood component transfusions comparing their compliance to NHS guidelines. The importance of this analysis can be explained first by the clinical point of view and second by the economic one.

## 1. Introduction

Blood transfusion represents a fundamental treatment of sick children, just as do pharmaceutical medications, intravenous fluids, and nutritional supplements. However, the misuse of blood transfusions can cause complications such as acute transfusion reactions or major allergic reactions [1,2].

In Italy, the blood transfusion network is handled by a public governance scheme. The National Blood Centre is the authority that coordinates and supervises the 21 regional blood transfusion centers to preserve homogeneous standards of quality and safety. This blood system is also aimed to reach national self-sufficiency for blood, blood components and plasma-derived medicines. The main difficulty in reaching this objective is that, in Italy, blood is collected only from voluntary and non-remunerated donors, making the blood supply determined by blood donor involvement [3].

In the context of pediatric blood transfusion, packed Red Blood Cells (pRBCs) are the most transfused blood component. pRBCs can be obtained via apheresis or by centrifugation of whole blood, or both [4]; while the other main blood components utilized in clinical activities are fresh frozen plasma (FFP), cryoprecipitate (Cryo), and platelets (PLT).

It is fundamental to remember that, despite substantial benefits for several diseases such as sickle cell disease, transfusions are associated with noninfectious and infectious risks (these are less and less frequent due to several donor screenings) [5,6,7,8,9,10,11,12]. In terms of adverse events, children are also likely at higher risk for noninfectious transfusion risks than adults [13,14]. Due to the risk without benefit increasing through blood transfusion, it is important to design blood management programs specifically for children, in addition to the fact that there is a notable lack of uniformity in transfusion practices and thresholds.

Despite blood transfusion’s importance in a Patient Blood Monitoring (PBM) program, few studies have assessed the actual current practices in pediatric hospitals or pediatric services [14]. A single retrospective cohort study compared the transfusion practices in hospitalized children in a large tertiary care hospital. This study highlights that RBC transfusion therapy varied significantly in hospitalized children for the 8 of 12 pediatric subspecialty services evaluated. This study concluded that transfusions may be overused in hospitalized pediatric patients. Based on this analysis it could be derived that a restrictive transfusion strategy could decrease the use of inappropriate RBC transfusions [15,16].

Despite the relevance of pediatric blood transfusions and their appropriateness, there are no specific international pediatric guidelines, only national disease-specific ones. Furthermore, the appropriateness of blood transfusion within hospital settings is largely unknown.

This study aims to assess the blood component use and appropriateness in pediatric wards of a research and teaching hospital, to develop internal instructions on the appropriate use and in the event to structure them as guidelines. During the analysis of clinical records, the blood component transfusions prescribed were compared with guidelines recommended by the British National Health Service (NHS) [17]. These guidelines are the only ones available in the literature that aren’t directly linked to a specific disease.

No ethical approval was needed for the study because the analysis of hospital medical records for quality, appropriateness and audit evaluation programs is allowed by law.

## 2. Materials and Methods

The research team performed a retrospective review of patients who were admitted to pediatric wards in a large tertiary hospital in Milan, The Fondazione Istituto di Ricovero e Cura a Carattere Scientifico Ca’ Granda Ospedale Maggiore Policlinico (hereafter the Fondazione).

The Fondazione has a pediatric emergency unit, a pediatric intensive care unit (PICUs) and a neonatal intensive care unit (NICUs), immunological, nutritional and infectious diseases pediatric wards, a cystic fibrosis center as well as a medical school, several post-graduate schools and, in addition, a few three-year courses for healthcare providers in the Faculty of Medicine and Surgery at the University of Milan. It also hosts a training center for postgraduate courses and first and second-level Master’s Degrees. Due to its organizational and clinical complexity, several challenging cases converge from all over Italy, creating a cohort of patients which needs a multidisciplinary approach that frequently cannot be treated according to guidelines.

Data were collected from patients that underwent at least a blood component transfusion from 2018 to the first six months of 2022. We assessed patients admitted to regular pediatric wards, involving chronic as well as acute pediatric diseases, in order to avoid misclassification due to critical situations that might require bypassing general pediatric blood transfusion guidelines. Transfusions performed in Pediatric Intensive Care or the Emergency Department were not analyzed. Furthermore, newborns were not considered. Data analyzed included: (1) patients’ characteristics; (2) the period of hospitalization and provision of blood component transfusion including the number of units transfused; (3) the category of transfused Fresh Frozen Plasma (FFP), Platelets (PLT) and Red Blood Cells (RBC) transfused; (4) and primary and secondary diagnoses at discharge.

The blood components analyzed were RBC, PLT and FFP, while data on blood component transfusion was provided by the Transfusion Centre Laboratory. The Fondazione’s Transfusion Centre Laboratory is part of the national transfusion center network and it is also a rare blood bank with 54 rare blood components as well as 30,763 blood components provided to clinical wards in 2021.

It should be noted that Fondazione’s Transfusion Centre Laboratory is arranged to prepare and deliver not only RBC packs but also pediatric RBC packs (defined as a Pedi-pack) to supply the right volume calibrated to the patients’ weight and to avoid a the misuse or waste of blood components. Pedi-packs could be required in the same way they are utilized to claim RBC to adult patients, detailing the volume required.

To define a patient’s characteristics and the blood components transfused feature, information was gained from several hospital information systems:

EMONET (Insiel Mercato SpA, Trieste, Italy): a laboratory information system that runs the whole workflow of blood component units from the blood collection of the donor through the Transfusion Centre to the transfusion to the patient. This information system is broadly used in Italy and is the reference system for the Regional Health Authorities.

CONCERTO (Dedalus Healthcare Systems Group SpA, Firenze, Italy): a laboratory information system that runs the whole workflow of a single laboratory test from the medical request (internal or external to the hospital) to the laboratory report. Some of the variables collected are defined by the Ministry of Health and the Regional Health Authority for systematic data collection for economic purposes (hospital reimbursement). The analysis was conducted by a multispecialty team formed by pediatricians and public health specialists to assess both the appropriateness and clinical fairness of prescription. Analysis was performed on clinical records files matching blood component requirements with clinical description and labor analysis results. During the prior analysis, the public health experts assessed the matching (as identified by NHS guidelines) between the blood transfusion prescription and the patient’s clinical features. Afterward, an analysis to assess the clinical features to explain the transfusion choice was performed by a multispecialist team of the Fondazione’s inner pediatricians, also involving pediatric hematologists and immunologists. The last phase consisted of a double review made by pediatric and public health experts to assess the pairing between clinical indication and hospital policy. The team was blinded about the identity of the prescriber. Finally, in cases of inappropriate prescriptions, we also investigated if any potentially related adverse event was recorded.

## 3. Results

From 2018 to the first six months of 2022, 5318 patients were admitted to the Fondazione’s pediatric wards. Table 1 reports the blood transfusion component in terms of patients’ demographic features, period of hospitalization and blood component category.

Table 2 reports data on the transfusion performed on the same patients through one or more hospitalizations and the assessment of transfusion performed during the same hospitalization.

After a first appropriateness analysis, only eight transfusions were in accordance with the NHS guidelines, with several transfusions performed under clinical specialist situations. As one of the most important Pediatric Nephrology centers in Italy, Fondazione is a point of reference for Hemolytic–uremic syndrome (HUS), which frequently requires blood transfusions. Table 3 reports the sets of diseases related to blood component transfusion. The hematologic genetic syndromes represent the group of diseases that require the largest number of blood components, with 41 (27.9%) components transfused, due to the several transfusions that every single patient required during hospitalization, followed by autoimmune diseases 31 (21.1%), owing to all the hemorrhages, such as skin or mucosal hemorrhages, caused by keratinization or collage formation failure. Acute anemia represents the third highest group of diseases in terms of the amount of transfused blood required, with 21 (14.3%) blood components.

Patients with HUS necessitated 17 (11.6%) transfusions, while 12 (8.2%) blood components were transfused due to the onset of Sickle Cell Syndrome complications. Lastly, all the other diseases with lower frequencies required a total of 14 (9.5%) blood transfusions.

In supplementary material we have reported also two assess about NHS indication about transfusion and the agreement between them and the transfusion performed in our unit, Appendix A: NHS clinical indication for Blood Component Transfusion; Appendix A: Demographic features and agreement with NHS criteria of Blood transfusion performed (*n* = 147).

In Table 4 we have reported the assessment of blood component transfusions and their appropriateness. Of approximately 147 components, only eight (5.4%) were transfused following the NHS indications, while 98 (66.7%) transfusions were led by the expertise of a clinician in a specific disease.

Furthermore, 39 (26.5%) transfusions were to be considered incorrect due to the absence of a clinical explanation or due to the absence of a specialist indication, such as an immunological or hematological indication.

Lastly, two (1.4%) blood components are material concerns due to the problem of transfusion tracking, despite blood packages being delivered to the operating theatre.

No adverse events potentially associated to inappropriate prescriptions were found.

## 4. Discussion

Blood transfusions are life-saving treatments that require significant attention due to the increasing lack of blood supply also linked to the lack of volunteer donors in Italy [18]. This problematic blood transfusion appropriateness is a major topic in healthcare management. Despite the presence of several guidelines for adult patients, a limited number of indications is available for pediatric patients. Linked to this, one of the main issues is the lack of specific thresholds to supply blood components. NHS guidelines show that blood transfusion needs to be carried out considering highly restrictive clinical conditions in order to avoid over-transfusion and infectious complications. [17].

Indeed, in our experience only eight (5.4%) transfusions were performed according to NHS guidelines, especially considering that six of them were performed for Sickle Cell Syndrome (SCS) with Acute Chest Syndrome that could be considered a clinical emergency. The strictness of guidelines about Sickle Cell Syndrome drives the inability to take as appropriate any transfusion performed on this syndrome despite the evidence of specific clinical manifestations or due to infections in the patients affected.

The absence of international guidelines or specific thresholds increases the challenge of the proper use of blood components, frequently delegating decisions to clinicians. The Fondazione’s pediatric wards are structured as a strict network made by various professionals to face rare diseases or unusual manifestations of common diseases. As previously described, Fondazione is a point of reference for several immunological or hematological conditions and rare diseases such as HUS and Epidermolysis Bullosa (EB); EB is a inheritable immunological disease that can lead to hemorrhaging due to skin fragilities or blister formation and subsequent rupture, spontaneous or iatrogenic rupture [19,20]. The chronic blood loss through the skin and the inadequate absorption of iron through intestinal mucosae could explain the chronic anemia that these patients suffered. At the same time, HUS, the main clinical disease to require RBC and FPP transfusions in our experience, is a group of blood disorders caused by infections and genetic mutations. Thrombotic Microangiopathy (TMA) is the common pathogenetic pathway leading to all forms of HUS. Due to the TMA, thrombocytopenia can be extreme, but despite this, platelet transfusions are not indicated unless there is evidence of active bleeding or specific genetic defects [21,22,23]. Indeed, transfused platelets can cause thrombotic events in various microvasculature beds of affected patients [24]. At the same time, RBC transfusions are considered fundamental in terms of supportive therapy in HUS and have been reported to be required in 80% of patients, due to the risk of severe anemia secondary to hemolysis, which can further destabilize patients by negative effects on the cardiovascular and respiratory systems [25,26]. The presence of a pediatric nephrology ward with a decade of experience in HUS management enables the assessment of the cause of HUS cause and the understand of the usefulness of blood transfusions compared to support therapy only.

This explains the rate of clinical decision mediated transfusions performed in our hospital. Despite the absence of specific indications provided by NHS guidelines, the decade-long experience of our pediatric nephrologists could provide enough expertise to assess the clinical manifestation and characteristics to appropriately choose the necessity of 16 (10.9%) blood transfusions. However, speaking about the genetic syndrome, their low frequency in the general population explains the lack of consideration made up by non-disease specific guidelines which force physicians to require blood components based on a combination of clinical signs, Hb levels, and experience.

At the same time, in all diseases, the third most common conditions are hemolytic anemias due to various chronic conditions, Glucose-6-phosphate dehydrogenase deficiency (G6PDD), or hereditary spherocytosis (HS). Because of the great complexity of pathophysiological processes involved in hemolytic anemia, there are no one-way direction treatments, but several guidelines and scientific society opinions. Various clinical approaches have been developed to address chronic anemia including iron supplies and the use of erythropoietin, but the main treatment remains blood transfusions of RBC. In this regard, especially in the absence of a unified recommendation presented with several different guidelines, pediatricians sometimes considered an HB level of 8 g/dL as a threshold to transfuse patients [27,28]. At the same time, several studies do not recommend transfusion solely on the basis of HB threshold, but highlight the importance of considering the occurrence of major bleeding or the presence of anemia symptoms [17,29,30].

For these reasons, we have considered as inappropriate some transfusions (approximately 25%) performed exclusively on the basis of low Hb or Plt levels.

It is interesting to note that these transfusions (*n* = 47, 32%) were performed during the first COVID wave. This might be explained by the fact that patients with rare diseases which could require transfusions were directed to our pediatric center.

It’s interesting to note that, despite the importance of the correct use of blood derivatives, the literature is poor in terms of appropriateness studies, while great attention will be paid to the outcomes of blood transfusions in patients that underwent surgery or who require several transfusions due a hematologic disease [31,32,33]. Due to the risk of bleeding, great attention was paid to post-operative transfusion, particularly in pediatric heart surgery, despite NHS guidelines suggesting the use of cryoprecipitate or the use of blood components only in specific clinical conditions, in interventions that could lead to massive bleeding, or in the event of a massive hemorrhage [17]. Despite unnecessary use of the blood supply or transfusion, the importance of blood as a lifesaving treatment and the necessity to preserve something that cannot be reproduced in a laboratory is also underlined by the large number of studies that try to deploy strategies to minimize the blood components used in clinical practice, such as by creating specific pediatric block-packs, known as pedi-packs, or by using a multi-disciplinary approach to achieve a reduction in blood products used for a single patient, as structured inside the Fondazione [34,35,36].

Lastly, two blood packs were difficult to track. They were sent to the ward but there are not any memos in the clinical records concerning them. One of them resulted in a patient being transfused but this only appears in the trasfusional center records and the other one is missed. Moreover, the above-mentioned importance of blood supplies underlines the importance of avoiding the waste of blood components and the difficulties faced in tracking two blood packs which were delivered to the clinical ward but were unmarked. These represent extremely important issues that underline the necessity to deploy a management and control system.

This study suffered from several limitations. For instance, it involved a relatively small sample size and was based on a single hospital. Furthermore, we have considered only the pediatric recovery ward without considering the Pediatric Emergency Department (PED) and Neonatal Intensive Care Unit or Pediatric Intensive Care Unit (NICU or PICU) due to the critical conditions of patients, which could have distorted the results.

## 5. Conclusions

Despite the importance of blood components and the growing difficulty to ensure their supply due to the reliance on volunteer donors, there are still no international guidelines that could lead an appropriate blood use. Some countries have developed specific guidelines which often assess a limited amount of disease to suggest blood component use. This reveals the range of prescription fairness, particularly in diseases that, although rare in the population, could be centralized to reference hospitals that may face problems of prescription appropriateness. Our analysis is one of the first appropriateness analyses performed to assess the accuracy of clinical indications for blood component transfusion comparing their compliance to the NHS guidelines. The importance of an appropriateness analysis could be explained in a two layer register view, the first one, being the clinical point of view, to assess and modify the previous existing guidelines based on a large series of rare diseases, and the second one the economic point of view, designed to improve a cost-effective policy due to the risk of waste a valuable life-saving treatment and the necessity to avoid this risk due to volunteer donors steadily decreasing. The analysis has shown that for specific clinical conditions physician expertise or specialist guidelines override blood transfusion guidelines when these do not consider specific illnesses.

Due to the importance of this topic and the lack of a match between NHS guidelines and several transfusion requirements, this experience could lead to a team of hematologists and pediatricians working to structure an internal procedure that could join specific thresholds and clinical signs, such as hemodynamic instability or minimum HbG thresholds, to infuse patients, especially in those clinical situations that, because of their rarity, could be not included in specific guidelines. 

## Figures and Tables

**Table 1 ijerph-20-01700-t001:** Feature of the blood component transfusion (*n* = 147) performed between 2018–2022.

Features of Transfusion	*n* (%)
Total blood component transfused	147
Total patients transfused	63
Gender *n* (%)	
M	54 (36.7%)
F	93 (63.3%)
Age	
Median, IQR	12.11
0–5	45 (30.6%)
5–10	20 (13.6%)
10–15	41 (27.9%)
15–30	41 (27.9%)
Period of hospitalization	
2018	37 (25.2%)
2019	31 (21.1%)
2020	47 (32.0%)
2021	23 (15.6%)
2022	9 (6.1%)
Blood component transfused	
FFP	37 (25%)
PLT	5 (3.4%)
RBC	105 (71.6%)

**Table 2 ijerph-20-01700-t002:** Number of blood transfusions to a single patient during one hospitalization (*n* = 147) or in different encounters (Np = 77) between 2018–2022.

**Transfusions performed during the same hospitalization**	***n* (%)**
One transfusion	46 (59.7%)
Two transfusions	24 (31.2%)
Three transfusions	3 (3.9%)
Four transfusions	2 (2.6%)
Nine transfusions	1 (1.3%)
Twenty-seven transfusions	1 (1.3%)
**Transfusions performed in different encounters**	**Np (%)**
One transfusion	35 (55.6%)
Two transfusions	19 (30.2%)
Three transfusions	2 (3.2%)
Four transfusions	1 (1.6%)
Six transfusions	3 (4.8%)
Nine transfusions	2 (3.2%)
Twenty-eight transfusions	1 (1.6%)

**Table 3 ijerph-20-01700-t003:** Diseases related to blood component transfusion (*n* = 147) performed between 2018–2022.

Clinical Conditions	*n* (%)
*Acute diseases*	38 (25.9%)
Hemolytic-Uremic Syndrome (HUS)	17 (11.6%)
Acute anemia	21 (14.3%)
*Exacerbation/complications of or new condition in Chronic diseases*	109 (74.1%)
Sickle Cell Syndrome (SCS)	12 (8.2%)
Neoplasms	11 (7.4%)
Hematological genetic syndromes	41 (27.9%)
Autoimmune diseases	31 (21.1%)
Other	14 (9.5%)

**Table 4 ijerph-20-01700-t004:** Appropriateness of blood component transfusions (*n* = 147) performed between 2018–2022 in accordance with NHS guidelines.

Appropriateness of Prescriptions	*n* (%)
*Appropriate*	8 (5.4%)
Sickle Cell Syndrome (SCS) with Acute Chest Syndrome	6 (4.1%)
Pancytopenia due to chemotherapy	2 (1.3%)
Hemolytic-Uremic Syndrome (HUS)	0 (0%)
Viral infections	0 (0%)
Anemia in genetic syndromes	0 (0%)
Hemolitic anemia	0 (0%)
Autoimmune diseases	0 (0%)
Infection in Sick Cell Syndrome (SCS)	0 (0%)
Other	0 (0%)
*Clinical Decision Mediated*	98 (66.7%)
Sick Cell Syndrome (SCS) with Acute chest Syndrome	0 (0%)
Pancytopenia due to chemotherapy	0 (0%)
Hemolytic-Uremic Syndrome (HUS)	16 (10.9%)
Viral infections	5 (3.4%)
Anemia in genetic syndromes	30 (20,5%)
Hemolytic anemia	4 (2.7%)
Autoimmune diseases	20 (13.6%)
Infection in Sick Cell Syndrome (SCS)	4 (2.7%)
Other	19 (12.9%)
*Inappropriate*	39 (26.5%)
Sick Cell Syndrome (SCS) with Acute chest Syndrome	4 (2.7%)
Pancytopenia due to chemotherapy	3 (2.0%)
Hemolytic-Uremic Syndrome (HUS)	1 (0.7%)
Viral infections	2 (1.4%)
Anemia in genetic syndromes	2 (1.4%)
Hemolytic anemia	1 (0.7%)
Autoimmune diseases	13 (8.9%)
Infection in Sick Cell Syndrome (SCS)	3 (2.0%)
Other	10 (6.8%)
*Not analyzable*	2 (1.4%)
*Required by anesthesiologists before surgery*	2 (1.4%)

## Data Availability

Data in Results section are internal data provided by internal transfusion laboratory service of Fondazione Istituto di Ricovero e Cura a Carattere Scientifico Ca’ Granda Ospedale Maggiore Policlinico.

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
