# Peer review of "Evaluation of Six Years of Appropriateness Level of Blood Transfusion in a Pediatric Ward"

_ijerph, 2023, doi:10.3390/ijerph20031700_

Round 1

Reviewer 1 Report

Dear Authors,

Your paper deals with a very important subject. You will find my comments below.

Some words already defined by an abbreviation are repeated as well as the description of the abbreviation (for example : RBC, FFP, PLT, ...).

In the Methodology, you describe the different services included in the hospital where the study was carried out. But you do not clearly explain that the study does not take account patients admitted to intensive care, emergency and neonatology (wich we learn in the discussion). This is needs to be clarified. 

Are the pediatricians who reviewed the records part of your hospital or do they come from other hospitals? This must be specified.

As there are more transfusion episodes than patients, it would be interesting to know if the multiple transfusions in a patient where administered during the same hospitalization or during different hospitalizations (and therefore to know the number of hospitalizations concerned by the transfused patients).

As the NHS guidelines are very dense, it would be interesting to summarize each criterion me or no met for each patient in a table (to put in supplementary material).

In methodology, you discussed about patients admitted in pediatric wards. However, in Results almost 28% of patients had 15-30 years. Can you comment on this ? in Itay, pediatric wards can accept patients up to what age?

One third of the transfusions took place during the first year of the COVID pandemic. Do you think that the pandemic and the fear of this new and unknown condition could affect transfusion decisions/thresholds by overworked doctors at the start of the pandemic? This should be discussed.

Author Response

Dear Authors,

Your paper deals with a very important subject. You will find my comments below.

Some words already defined by an abbreviation are repeated as well as the description of the abbreviation (for example : RBC, FFP, PLT, ...).

Authors’ response: Thank you for this remark. We have edited the manuscript according to this suggestion

In the Methodology, you describe the different services included in the hospital where the study was carried out. But you do not clearly explain that the study does not take account patients admitted to intensive care, emergency and neonatology (wich we learn in the discussion). This is needs to be clarified. 

Authors’ response: We thank the reviewer for this pertinent commet. In the revised version of the manuscript, we provide a clearer description of pediatric wards involved in the study and why we have choose these specific settings.

Are the pediatricians who reviewed the records part of your hospital or do they come from other hospitals? This must be specified.

Authors’ response: We clarified this important point specifying that pediatricians were inner foundation pediatricians

As there are more transfusion episodes than patients, it would be interesting to know if the multiple transfusions in a patient where administered during the same hospitalization or during different hospitalizations (and therefore to know the number of hospitalizations concerned by the transfused patients).

Authors’ response: We found this comment interesting. We have introduce a new table with blood transfusions performed during the same hospitalization and blood transfusions performed for single patient in the revised version of the manuscript

As the NHS guidelines are very dense, it would be interesting to summarize each criterion me or no met for each patient in a table (to put in supplementary material).

Authors’ response: Thank you for this remark. We have edited the required table

In methodology, you discussed about patients admitted in pediatric wards. However, in Results almost 28% of patients had 15-30 years. Can you comment on this ? in Itay, pediatric wards can accept patients up to what age?

Authors’ response: Thank you for this question . This population is followed by inner Fondazione pediatricians to a specific rare disease that due the high expertise and the specific disease clinical knowledge of our physician required a strictly connection even after the pediatric age limit. As in many other centers in Europe (PMID: 35084548).

One third of the transfusions took place during the first year of the COVID pandemic. Do you think that the pandemic and the fear of this new and unknown condition could affect transfusion decisions/thresholds by overworked doctors at the start of the pandemic? This should be discussed.

Authors’ response: Thank you for this question. We have addressed this issue in the revised version of the discussion

Reviewer 2 Report

The authors evaluated appriopriatness of transfusions in pediatric wards in a large tertiary hospital of Milan in years 2018-2022 according to British National Health Service guidelines recommendations. There were 147 transfusions evaluated. Here are some questions arised while reading: according to table 3 appropriated prescriptions were in 8 cases inappropriate in 39.Clinical decision mediated were in 98 cases and my question is if they were prescribed properly to NHS guidelines?

In part of inappropriated transfusion what was wrong, comment?

If there were inappropriate transfusions were any complications arised?

Lines 238 - 245 not understandable.

Is there any difference in treatment of patients of UK and Italy and transfusions proscribed?

Author Response

The authors evaluated appriopriatness of transfusions in pediatric wards in a large tertiary hospital of Milan in years 2018-2022 according to British National Health Service guidelines recommendations. There were 147 transfusions evaluated. Here are some questions arised while reading: according to table 3 appropriated prescriptions were in 8 cases inappropriate in 39.Clinical decision mediated were in 98 cases and my question is if they were prescribed properly to NHS guidelines?

Authors’ response: Thank you for this comment. The 98 cases could be considered appropriate according to NHS guidelines, we differentiate them considering the clinical description and rationale made up by pediatricians, while 39 inappropriate did not show a clinical rationale except for low Hb levels.

In part of inappropriate transfusion what was wrong, comment?

Authors’ response: Inappropriate transfusions were driven exclusively by hemoglobin levels without considering clinical records or clinical history and considering adult thresholds.

If there were inappropriate transfusions were any complications arised?

Authors’ response: In our experience there were no complications due to these 147 transfusions, directly linked to blood transfusion. We added this information in the revised version of the manuscript

Lines 238 - 245 not understandable.

Authors’ response:  We apologize for that. We have rewritten these sentences to make them clearer.

Is there any difference in treatment of patients of UK and Italy and transfusions proscribed?

Authors’ response:  The question is relevant and interesting, but we cannot fully address it due to the absence of specific Italian guidelines.